# Use of telemedicine in general practice in Europe since the COVID-19 pandemic: A scoping review of patient and practitioner perspectives

**David Walley**[1], **Geoff McCombe**[1]*, **John Broughan**[1], **Conor O'Shea**[2], **Des Crowley**[1,2], **Diarmuid Quinlan**[2], **Catherine Wann**[3], **Tadhg Crowley**[4], **Walter Cullen**[1]

**1** UCD School of Medicine, University College Dublin, Dublin 4, Ireland, **2** Irish College of General Practitioners, Dublin, Ireland, **3** Nobber Health Centre, Nobber, Ireland, **4** Ayrfield Medical Practice, Kilkenny, Ireland

* geoff.mccombe@ucd.ie

## Abstract

General practice is generally the first point of contact for patients presenting with COVID-19. Since the start of the COVID-19 pandemic general practitioners (GPs) across Europe have had to adopt to using telemedicine consultations in order to minimise the number of social contacts made. GPs had to balance two needs: preventing the spread of COVID-19, while providing their patients with regular care for other health issues. The aim of this study was to conduct a scoping review of the literature examining the use of telemedicine for delivering routine general practice care since the start of the pandemic from the perspectives of patients and practitioners. The six-stage framework developed by Arksey and O'Malley, with recommendations by Levac et al was used to review the existing literature. The study selection process was conducted according to the PRISMA Extension for Scoping Reviews guidelines. Braun and Clarke's 'Thematic Analysis' approach was used to interpret data. A total of eighteen studies across nine countries were included in the review. Thirteen studies explored the practitioner perspective of the use of telemedicine in general practice since the COVID-19 pandemic, while five studies looked at the patient perspective. The types of studies included were: qualitative studies, literature reviews, a systematic review, observational studies, quantitative studies, Critical incident technique study, and surveys employing both closed and open styled questions. Key themes identified related to the patient/ practitioner experience and knowledge of using telemedicine, patient/ practitioner levels of satisfaction, GP collaboration, nature of workload, and suitability of consultations for telemedicine. The nature of general practice was radically changed during the COVID-19 pandemic. Certain patient groups and areas of clinical and administrative work were identified as having performed well, if not better, by using telemedicine. Our findings suggest a level of acceptability and satisfaction of telemedicine by GPs and patients during the pandemic; however, further research is warranted in this area.

**Data Availability Statement:** This study is a scoping review of the literature on our study topic. As such, data for this study is the contents of the

published articles included in this review. The list of articles reviewed in this manuscript can be accessed via Table 1 and the reference list. Table 1 also outlines specific details of the data that was extracted from the included articles for the purpose of this study.

**Funding:** We are grateful to the Ireland East Hospital Group, UCD College of Health & Agricultural Sciences and UCD School of Medicine who funded the salaries of authors GM and JB for the duration of their work on the study. The funders had no role in study design, data collection and analysis, decision to publish, or preparation of the manuscript.

**Competing interests:** The authors have declared that no competing interests exist.

## Author summary

Globally, the COVID-19 pandemic imposed significant restrictions on social contact. These included practices such as social distancing and cocooning for elderly people and those with various morbidities. This resulted in general practitioners (GPs) across Europe having to adopt to using telemedicine consultations to minimise the number of social contacts made. GPs had to try to achieve a balance between preventing the spread of COVID-19 and providing their patients with regular care for other health issues. In this paper we conducted a comprehensive review of the literature to examine the use of telemedicine for delivering routine general practice care since the start of the pandemic from the perspectives of patients and practitioners. Our findings suggest a level of acceptability and satisfaction of telemedicine by GPs and patients during the pandemic. However, for doctors to be prepared to make the shift to telemedicine, they require training and education on using telemedicine as well as ensuring they are equipped with the necessary digital resources to conduct remote care.

## Background

Attempts to manage the COVID-19 pandemic have led to radical global reorganisations of society and health care systems. [1] General practitioners (GPs) are the first point of contact for patients with health concerns, and they provide the vast majority of patient care and treatment. Therefore, during the pandemic GPs had to balance two needs: preventing the spread of COVID-19, while providing their patients with regular care for other health issues. This has led to a greater use of telemedicine and remote consulting to treat patients since the start of the COVID-19 pandemic.

There is no single definition of telemedicine on which the majority of people agree; however, the general recurring terms concur that telemedicine refers to the use of telecommunications technology in the remote diagnosis, treatment, and care of a patient. [2] Such technologies include mobile phones, tablets and laptops, where consultations are carried out by means of a text messaging, telephone or video call. The World Health Organisations promotes the use of telemedicine for research and education purposes, stating that telemedicine is an 'open and constantly evolving science'. [3] Remote consulting is another term used to describe the way in which practitioners deal with patients online at a distance. The meaning of this term is limited to the individual consultation process itself, and does refer to telemedicine on a systematic basis, or its use for educational purposes. [4]

Globally, the availability and use of telemedicine differs between countries. Wealthier countries, and those with higher rates of Gross Domestic Product (GDP) spent on health, consume greater amounts of telemedicine. [5] Bigger countries with large populations living in remote areas have higher usage of telemedicine. [6] Prior to the COVID-19 pandemic, most telemedicine publications relate to the necessity for telemedicine to tackle the issue of access to healthcare for rural populations. Currently, the main issue for which telemedicine is being used is to reduce the transmission of COVID-19.

In March 2020, Europe was declared as the epicentre for COVID-19. [7] At this time, The European Centre for Disease Prevention and Control (ECDC) reported that the then-current mitigation measures were not sufficient to suppress the virus, and that further mitigation strategies were needed in order to reduce burdens on national health services. [8] By mid-March 2020, 250 million Europeans were in lockdown, [9] as public health measures were introduced to close non-essential business and services in order to limit person-to-person contact.

As general practice is the first point of contact for patients, it was responsible for dealing with patients with COVID-19. [10] Across Europe, GPs were responsible for referring patients for tests for COVID-19, diagnosing COVID-19, advising patients on self-isolation methods, informing patients of respective public health measures, and referring patients to hospital. [11] General practice continued exercising existing infection control measures, whilst implementing new advice on infection control from ECDC including using personal protective equipment (PPE) in clinical settings when caring for patients. [12] PPE consisted of eye protection (goggles and face shield), respiratory protection (face mask), and bodily protection (gloves and gowns). [13] Both the numbers of cases of COVID-19, as well as public health measures differed across countries in Europe. [14]

In Ireland, the use of telemedicine since March 2020 increased by a multiple of five, with 20% of the population reporting to have interacted with the health service through telemedicine. [15] Telehealth has been an important way in which health services have adapted to care delivery in a pandemic and exploiting its potential in healthcare delivery had also been a priority. In Ireland, health policy recognises how telemedicine can be used to reduce burdens on the health service, [16] however, it is not predicted that it will replace traditional in-person visits, but instead act as an ancillary tool. [17] Current government policy aims to increase the number of consultations carried out remotely by investing in relevant infrastructure and aims to educate patients on how they can use telemedical services. It also recognises the need for research in order to achieve the best outcomes. [18]

In general practice, while extensive literature examining telemedicine exists, relatively little literature examining the issue since the pandemic exists. This is especially the case where 'routine' GP care is concerned. We sought to address this knowledge gap by examining the use of telemedicine for delivering routine GP care since the pandemic and especially literature examining the perspectives of patients and practitioners.

## Methodology

The chosen methodology was a scoping review. This scoping review was conducted between June and September 2021 and followed the six-stage process outlined by Arksey & O'Malley to collate existing literature, identify key findings and outline current research gaps in this area. [19]

**Stage 1: Identifying the research question.** The COVID-19 pandemic has had huge impacts across all areas of society in European countries. Likewise, its impact on health services has been significant, particularly on general practice. In order to fulfil public health social distancing regulations, general practice implemented telemedicine, such as teleconsultations and remote prescribing. It is important to understand this unprecedented change from both the patients' and practitioners' perspectives. Hence, the objective of this scoping review was to examine recent literature relating to the use of telemedicine in general practice in Europe since the start of the COVID-19 pandemic. The following question was formulated: Patient/ Practitioner perspectives of remote consulting in primary care in Europe since the COVID-19 pandemic.

**Stage 2: Identifying relevant studies.** A preliminary search of key databases and the grey literature was performed, using multiple search terms to create a reading list. From this, keywords were identified and medical subject heading (MeSH) terms were generated (Fig 1). The electronic databases used in the literature search were PubMed, Cochrane Library, Cinahl and Embase. Additionally, studies were added by hand-searching. In order to focus on the COVID-19 pandemic literature, the search was limited to publications from 2020 onwards and was limited to publications set in European contexts.

(((((Primary Care)[Title/Abstract] OR (General Practice)[Title/Abstract] OR (Family Medicine)[Title/Abstract] OR (GP)[Title/Abstract]) AND ((Remote Consult*)[Title/Abstract] OR (Online Consult*)[Title/Abstract] OR (Telemedicine)[Title/Abstract] OR (Telehealth)[Title/Abstract])) AND ((COVID-19)[Title/Abstract] OR (Coronavirus)[Title/Abstract] OR (Pandemic)[Title/Abstract] OR (COVID)[Title/Abstract])) NOT ((Hospital)[Title/Abstract])) NOT ((outpatient)[Title/Abstract])

**Fig 1. Search strategy.**

**Stage 3: Selecting studies.** The initial search generated 797 studies, which were compiled into a preliminary reading list. This included nine hand-selected studies. The selection process consisted of a review of titles, and abstracts, followed by full-text reviews. The 'Preferred Reporting Items for Systematic Reviews and Meta-Analyses (PRISMA)' flow diagram below (Fig 2) outlines the selection process. Upon completion of the search, 125 duplicates were removed. EndNote 20 software was used to track and group studies, manage citations and remove duplicates. Studies were included if they were considered to examine the research question, if they were published in English and if the full text article was available. Findings

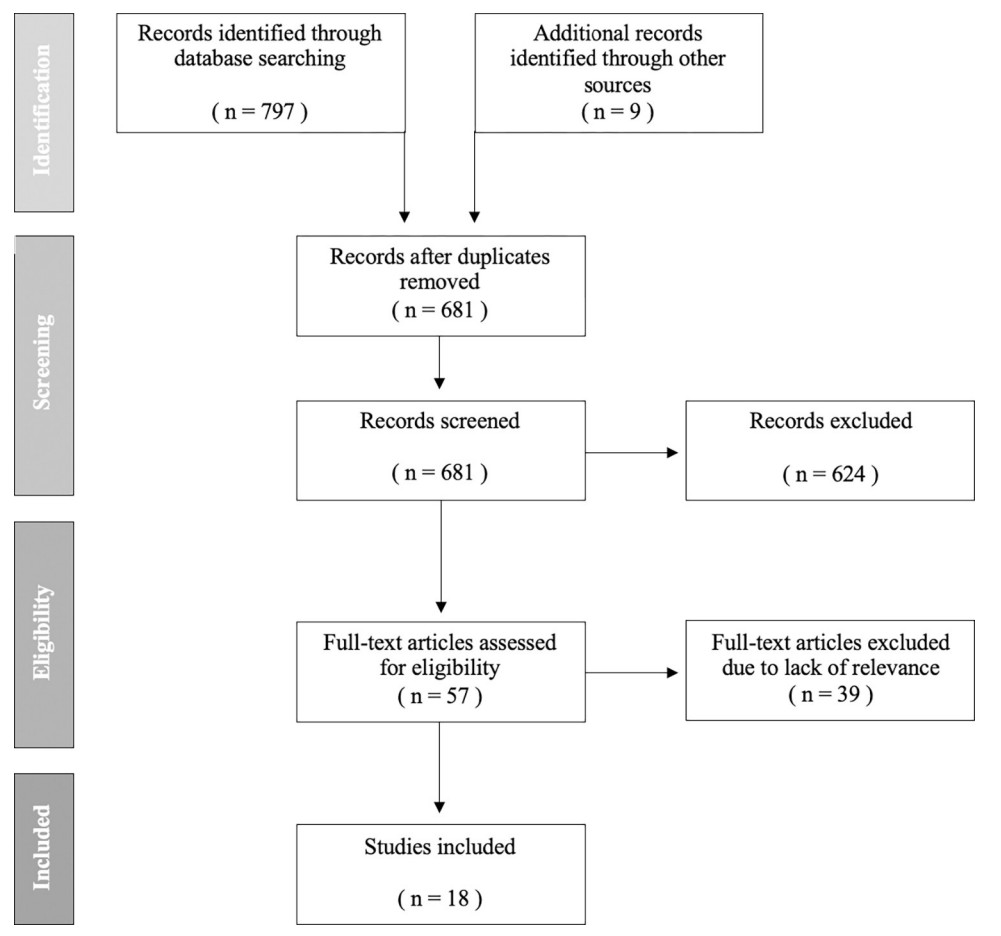

**Fig 2. PRISMA Flow Diagram.**

were reviewed by a second reviewer, and a finalised list of studies was agreed. Studies were included if the following inclusion criteria applied:

- Published from 2020 onwards

- Set in General Practice

- Set in Europe

- Available in English

- Provided patient/ practitioner perspective

- Full text article available

- Not an opinion piece or editorial

**Stage 4: Charting the data.** Once all relevant articles were identified, to facilitate comparison and thematic analysis, the following data was charted from the articles:

- Author(s), Year of Publication

- Title

- Journal/Publication

- Location

- Population

- Aim/topic

- Study design

- Major Finding

**Stage 5: Collating, summarising and reporting results.** An overview of the literature is detailed in Table 1 below, summarising and charting the results. This will be discussed further in the results section.

**Stage 6: Consultation.** As per the recommendations regarding scoping reviews by Levac et al., [20] a consultation was conducted with experts in the area of telemedicine in general practice. Studies were included/ excluded based on their advice.

## Results

The initial search generated 797 studies and a further nine studies were sourced through hand searching. After 125 duplicates were removed, reviewers screened the remaining 681 studies by title and abstract, during which 624 studies were excluded. Fifty-seven studies met the inclusion criteria and were selected for full-text review. Following full-text review, 39 studies were excluded due to a lack of relevance or unavailability of the full-text, leaving 18 studies which examined the use of telemedicine in general practice in Europe since the start of the COVID-19 pandemic.

The included studies were from nine different countries. Seven studies were set in the United Kingdom, whilst other countries only have one relevant study included in this review. Thirteen studies explore the practitioner perspective of telemedicine, while five studies looked at the patient perspective of telemedicine. The types of studies which were included in this review were: qualitative studies, literature reviews, systematic review, observational studies,

**Table 1.  Summary of included studies.**

| Author, Year | Journal | Location | Population | Aim/ Topic | Study Design | Major Finding |
|---|---|---|---|---|---|---|
| Archer, S et al. (2021) [41] | British Journal of General Practice | UK | 23 GPs | Impact of COVID on cancer assessment in primary care | Qualitative interview | - Inability to assess subtle symptoms online <br> - Reported more risk taking in care |
| Byrce, C et al. (2021) [42] | British Journal of General Practice | UK | 2789 Patients | Tele-access to GP | Self-administered survey | Strong association between tele-knowledge and usage |
| Danhieux, K et al. (2020) [32] | BMC Family Practice | Belgium | 16 Primary Care Centres | Provision of chronic care | Qualitative study | Attitudes + concerns of GP |
| Due, T.D (2021) [23] | BMC Family Practice | Denmark | 13 GPs | GP use of alternative consult | Qualitative interviews | Found telemedicine useful |
| Duncan, LJ et al. (2021) [28] | F1000 Research | UK | 150 patients | Perception of NHS general practice during pandemic | 2 online surveys | Varied result–finely split. Opinion differs re. access, quality |
| Eisele, M et al. (2021) [27] | Frontiers in Medicine (Lausanne) | Germany | 121 GPs | Examine general challenges faced by GPs during pandemic | Open questions survey | GP payment (no financial worry) |
| Florea M et al. (2021) [33] | Int. J. General Medicine | Romania | 108 GPs | Perception of Telemedicine | Cross section study | GPs received positive feedback from patients on telemedicine |
| Johnsen et al. (2021) [35] | Journal of Medical Internet Research | Norway | 1237 GPs | Document GPs experience of video consultations during pandemic | Cross-sectional survey | GPs lacked prior experience of VC Consultation suitability greatly varied across patient groups GPs estimate 20% of future work will be conducted online |
| Kurotschka, P et al. (2021) [31] | Frontiers in Public Health | Italy | 149 GPs | To explore Italian GPs care experience during 1st wave of COVID-19 | Critical incident technique study | GPs satisfied with remote consultation Despite not having physical sign/ presence, GPs found that verbal communication became more effective to suffice this |
| López Seguí, F et al. (2020) [43] | Journal of Medical Internet Research | Spain | 5382 Consultations 20 GPs | Evaluate whether a consultation is best suited to teleconsult/ in-person | Retrospective cross-sectional study | Teleconsultations conducted most often were: managing test results, repeat prescriptions, general medical enquiries |
| Murphy, M et al. (2021) [26] | British Journal of General Practice | UK | 87 Staff 350000 Patients reg. | Practitioner Perspective | Mixed-methods Longitudinal study Normalisation Process theory | Universal consensus re. necessity |
| Parker, R et al. (2021) [44] | British Journal of General Practice | UK | 13 studies | Explore impact of remote consult in general practice across socioeconomic groups | Systematic review | Remote consult more likely to be used by younger, non-immigrant, employed, female patients while online consult will be greater used by young affluent and educated groups |
| Saigí-Rubió, F et al. (2021) [29] | Journal of Medical Internet Research | Spain | 1189 Healthcare professionals | Study determinants to use e-consult platform | Qualitative/ Quantitative questionnaire | Discover determining factors |
| Saint-Lary, O et al. (2021) [25] | British Medical Journal | France | 7481 GPs | Change to French GP | Observational study | Widespread changes in French GP |

(*Continued*)

**Table 1.** (Continued)

| Author, Year | Journal | Location | Population | Aim/ Topic | Study Design | Major Finding |
|---|---|---|---|---|---|---|
| Solans, O et al. (2021) [34] | Journal of Medical Internet Research | Spain | 5.8 Million | Profile of eHealth users | Descriptive, Observational Study | General changes in profile across patient groups |
| Tuijt, R et al. (2020) [45] | British Journal of General Practice | UK | 30 Patients | Dementia patient attitude to remote consult | Qualitative interviews | Found calls reassuring but limited in scope |
| Verhoeven, V et al. (2020) [24] | British Medical Journal | Belgium | 132 GPs | Impact of COVID on primary care | Qualitative interview | Increased workload re. training + keeping up to date on knowledge re. COVID + online triage |
| Wanat, M et al. (2021) (22) | British Journal of General Practice | UK | n/a | Understand how dif. Euro countries PCPs dealt with first wave | Exploratory qualitative study Semi-structured interviews | Generally–dealt well. Variation across countries. |

quantitative studies, Critical incident technique study, and surveys employing both closed and open styled questions. The 18 included studies were thematically analysed as informed by Braun and Clarke [21] and the following six key themes were identified.

### Experience & knowledge of remote consultation

Six of the 13 practitioner related studies addressed the issue of GPs' experience of telemedicine, and their knowledge of the issue. GPs in countries such as the UK, Norway and Sweden reported having some experience of telemedicine, which made their transition to almost complete use of telemedicine in their clinic easier. [22] However, the majority of the studies stated that the practitioners involved had no experience of telemedicine. [23,24] GPs nonetheless employed the use of telemedicine in their practice. The main reason for moving to remote practice was in the interest of infection control. [25] GPs reportedly did not feel aggrieved by this change as they felt that it was in everyone's interest. [26] GPs reported that guidance regarding telemedicine from various bodies was often conflicting with one another, which made implementing telemedicine into practice more difficult. [27]

### Patient satisfaction

Three out of the six patient-related studies addressed the issue of patient satisfaction. In the UK, 78% of general practice patient were satisfied with the care they received and reported feeling confident that the care they received adequately address their health condition. [28] Patients reported that they were not willing to engage with general practice due to concerns of catching COVID-19; however, remote consultations alleviated this concern by removing any risk of infection. [28] Patients were also satisfied with the degree of convenience provided by telemedicine. [29] Another issue which affected patients' satisfaction was cost. Patients were not willing to spend as much on a teleconsultation as they would on an in-person consultation. [30]

### Collaboration

GPs in several studies highlighted how they engaged with other GPs, within their practice, and outside of their practice. They commented on how this was newly developed. Large practices developed teams to plan the type of care which will be conducted remotely. [26] Other large

practices divided out their workload so that one individual would carry out only telephone consultations, or only video consultations etc. This specialisation allowed for doctors to become used to the remote services, whilst also helping older or more vulnerable staff members to shield. [31]

## Workload

Six of the-practitioner based studies addressed the question of workload. There was a rather varied opinion within and between the studies included in this review. Although there was less demand for services, GPs reported an increase in workload as a result of telemedicine. This increase in workload arose from learning how to use telemedical services and maintaining a more comprehensive administrative record for teleconsultations. GPs reported a drop in chronic care consultations as a result of patient stratification. On the other hand, the same study reported that telemedicine helped to reduce the GPs' workload in relation to remote prescribing. [32] Another study reported a similar reduction in workload, however it also described a change in the nature of the practitioners work. Consultations became more targeted, less 'chit chat' involved, and were shorter in nature. The same study mentioned how GPs increased house calls in order to protect elderly from coming in contact with COVID-19 when travelling to their GP surgery. [23] GPs reported that care provided was generally similar to, if not better than, in-person consultations. One study reported that 51% of GPs thought the care they provided was similar to in-person care.

## Future of telemedicine

The future of telemedicine in general practice appeared uncertain, as there is very little consensus on the issue amongst doctors. Due to inexperience, doctors feel they are more comfortable continuing with in-person care once it is permitted (26). However, other GPs stated that they found telemedicine easy to conduct, and would estimate that 20% of their future consultations will be conducted remotely. [33]

## Suitability of consultation

The literature agreed as to the suitability of certain patients and consultation types for telemedicine. Telemedicine was more suited for younger, urban, more educated patients who are working. Adult female patients and pregnant women are also suited to telemedicine. [30,34] GPs reported that they had good experience of treating mental health consultations remotely, particularly through video consultations, and would be happy to continue to do so. [35] GPs reported that they were happy to prescribe remotely, as well as conduct general medical enquiries remotely. [32]

## Discussion

### Key findings

The authors identified 18 studies exploring practitioner and patient perspectives of telemedicine in primary care in Europe since the start of the COVID-19 pandemic. The findings indicate that there was a high level of satisfaction of using telemedicine amongst both patients and GPs. Practitioners were satisfied by the use of telemedicine as it minimised the risk of catching COVID-19 from patients which allowed them to serve their patients' needs during the pandemic. The findings also suggest that GPs collaborated with one another, and other external bodies in unprecedented ways during the pandemic. This included the sharing of resources like PPE, and the sharing of personal experience and specialist knowledge. Most GPs working

in Europe had not experienced working in a pandemic which suggests why doctors reached out to one another to collaborate. Doctors had to share knowledge and skills with one another, and discuss new information, in order to provide the highest standards of patient care. Types of care which doctors reported as being most suited to telemedicine tended to be mostly administrative, e.g. repeat prescriptions, or generally brief in nature, e.g. reporting test results, while physical examinations or procedures would require a face-to-face consultation.

The reason for the high level of satisfaction of telemedicine amongst patients is varied. Patients fear of catching COVID and wishing to adhere to public health advice were pleased that telemedicine enabled them to receive medical advice or routine care while meeting both of these needs. Younger patients were more suited to telemedicine as a great deal of their lives is conducted using technology. Many people who come into contact with computers and various IT interfaces at work adjusted well to using telemedicine to engage with their GP.

## Methodological challenges

Our study used a robust review methodology following the framework provided by Arksey and O'Malley. [36] However several limitations should be considered when interpreting the findings of this review. Whilst we adopted a rigorous scoping review methodology and used a comprehensive search approach, there is a possibility that not all publications relevant to the inclusion criteria were identified by the searches or databases used. Though by conducting stage six of the Arksey and O'Malley framework which involved consulting with experts in the field we aimed to minimise exclusion of any relevant studies. Furthermore, scoping reviews do not include an assessment of study quality as the focus is on covering the range of work that informs the topic rather than limiting the work to studies that meet particular standards of scientific rigour. Finally, only articles published in English were considered for inclusion in our review, which could have resulted in the exclusion of equally relevant literature published in other languages.

## Comparison with existing literature

Doraiswamy and colleagues undertook a review of telemedicine during COVID-19 across all healthcare settings globally. [37] Their research reported high levels of satisfaction amongst practitioners; however, the literature was mostly from the United States and were mostly opinions, commentaries and perspectives. Similar to our review, their research highlighted many aspects of clinical practice being conducted virtually to adhere to public health restrictions. The authors also believed that many aspects of telemedicine will remain after the COVID-19 pandemic.

A review based on the implementation and usefulness of telemedicine by Hincapié et al. suggested that both patients and practitioners had high levels of acceptability on the use of telemedicine. [38] This review found that telemedicine was particularly favoured by those who travel long distances to receive care. Although focussed on hospital out-patient and in-patient clinics, the study concurs with our findings that telemedicine was implemented in order to achieve continuity of care for patients. The study mentioned how different institutions communicated with one another to listen to expert guidance and counsel on medical and/ or administrative decisions. We found similar levels in communication in the studies included in this review.

Government policy, such as the Department of Health (Ireland), suggests that resources be spent on research to improve telehealth services, and to expand on telemedicine services post-pandemic due to their reported success. Target areas include vulnerable populations, telepsychiatry and virtual triage for emergency departments. [39] The findings of our study suggest

that telemedicine can help in the conduct of routine general practice post-pandemic, as it evidently fared well during the COVID-19 pandemic; however, more research is required to inform future clinical policy on telemedicine.

## Implication for research and practice

As indicated in the studies included in this review, GPs wish to maintain a level of telemedicine in their daily clinical practice. GPs highlighted certain aspects of clinical and administrative work that improved due to telemedicine, and they stated they would like this to continue beyond the pandemic. GPs highlighted their use of trial and error methods in approaching clinical issues through telemedicine. Going forward, there needs to be clarity as to what areas of primary care can be conducted effectively using telemedicine so that the outcomes are equal to, if not better than face-to-face consultations. Furthermore, Government policy across many countries is calling for increased use of telemedicine in healthcare. [40] Increased investments in post-pandemic healthcare budgets across Europe may act as a catalyst to the implementation of such health care policies. GPs must be prepared for such changes in their respective health services and as such clear guidance from training and professional bodies and medical council is required.

## Conclusion

Although the prevalence of COVID-19 has fallen in Europe due to successful vaccination programmes, doctors must be prepared to make the shift to telemedicine at short notice. It is evident that training and education on using telemedicine for doctors is needed, as well as ensuring they are equipped with the necessary digital resources to conduct remote care. Telemedicine is a relatively new aspect of primary care and more research is warranted to optimise its use in general practice. The impact of telemedicine on the quality of patient care and clinical outcomes is relatively unknown and requires further exploration.

## Supporting information

**S1 PRISMA Checklist. PRISMA Checklist.**
(DOCX)

## Author Contributions

**Conceptualization:** Geoff McCombe, Walter Cullen.

**Data curation:** David Walley.

**Formal analysis:** David Walley.

**Funding acquisition:** Walter Cullen.

**Methodology:** Geoff McCombe.

**Project administration:** Geoff McCombe, Walter Cullen.

**Supervision:** Geoff McCombe, John Broughan, Walter Cullen.

**Writing – original draft:** David Walley, Geoff McCombe.

**Writing – review & editing:** Geoff McCombe, John Broughan, Conor O'Shea, Des Crowley, Diarmuid Quinlan, Catherine Wann, Tadhg Crowley, Walter Cullen.

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
