## [Decision Letter · Decision Letter 0]

14 Mar 2023

PDIG-D-22-00363

Use of telemedicine in general practice in Europe since the COVID-19 pandemic: a scoping review of patient and practitioner perspectives.

PLOS Digital Health

Dear Dr. McCombe,

Thank you for submitting your manuscript to PLOS Digital Health. After careful consideration, we feel that it has merit but does not fully meet PLOS Digital Health's publication criteria as it currently stands. Therefore, we invite you to submit a revised version of the manuscript that addresses the points raised during the review process.

In particular, we invite you to address the following points: 

- provide further details about methods for the study, as requested by Reviewer 1 and Reviewer 3

- check (and amend if necessary) that all sources are cited and referenced correctly

- Reviewer 1 suggests the need for further reflection in the discussion on implications for practice and further research

Please submit your revised manuscript within 30 days Apr 13 2023 11:59PM. If you will need more time than this to complete your revisions, please reply to this message or contact the journal office at digitalhealth@plos.org. Please include the following items when submitting your revised manuscript:

We look forward to receiving your revised manuscript.

Kind regards,

Valentina Lichtner

Academic Editor

PLOS Digital Health

Journal Requirements:

Additional Editor Comments (if provided):

Reviewers' comments:

Reviewer's Responses to Questions

**Comments to the Author**

1. Does this manuscript meet PLOS Digital Health’s publication criteria? Is the manuscript technically sound, and do the data support the conclusions? The manuscript must describe methodologically and ethically rigorous research with conclusions that are appropriately drawn based on the data presented.

Reviewer #1: Yes

Reviewer #2: Yes

Reviewer #3: Partly

2. Has the statistical analysis been performed appropriately and rigorously?

Reviewer #1: N/A

Reviewer #2: Yes

Reviewer #3: N/A

3. Have the authors made all data underlying the findings in their manuscript fully available (please refer to the Data Availability Statement at the start of the manuscript PDF file)?

Reviewer #1: Yes

Reviewer #2: Yes

Reviewer #3: No

4. Is the manuscript presented in an intelligible fashion and written in standard English?

Reviewer #1: Yes

Reviewer #2: Yes

Reviewer #3: Yes

5. Review Comments to the Author

Reviewer #1: The authors are commended for addressing an important research question and summarising the findings of the evidence base. The paper is well written and structured, concise, and of adequate English language standard. The paper has merit, but would benefit from some minor revisions. The following comments are intended to improve the paper:

1. PRISMA has been updated in a 2020 Statement. Inclusion of the updated PRISMA flowchart could be considered for your paper. This will allow elaboration of minor search details that are lacking in the body of the text, including: specific reasons for article exclusion, number of full texts unable to be retrieved, number of grey literature vs hand selected articles.

2. Could you elaborate on the outcomes from the consultation with experts in the field and how this informed the inclusion of articles? i.e. who was consulted? what was this process? how many studies were included/excluded? Could you also clarify the purpose of ‘Step 6: Consultation’: was this to determine the articles’ quality or content related to the field of telemedicine?

3. Acknowledgement of not undertaking quality assessment of included studies in a scoping review is noted in the discussion section under methodological concern. The paper may be strengthened by explicitly cautioning the reader of interpretation of the findings based on the absence of a quality assessment of included studies.

4. The finding that some aspects of telemedicine in GP practice were perceived by GPs as equivalent to face-to-face consultations is of interest. Were there any findings from studies on patient perspectives regarding telemedicine vs face-to-face?

5. Patient satisfaction with telemedicine was high. Were there additional factors to costs affecting patient satisfaction of telemedicine?

6. Certain patient groups (young, urban, educated, working) were identified as being suitable to telemedicine. This may lead into the discussion about future research into barriers to using and accessing technologies for disadvantaged patient groups that would benefit from telemedicine.

7. As the world moves on from the COVID-19 pandemic, this paper will be an important reference on the immediate uptake on telemedicine in GP practice during the pandemic. Could you comment on what this paper’s findings/ themes say about the sustainability of telemedicine in clinical practice. What factors are important for telemedicine’s ongoing use, such that clinical practice does not revert back to pre-pandemic ways of working?

Reviewer #2: Thank you for this manuscript.

Please check all your references if they have the correct numbering in the text. For example, on page 6 of the paper, line 121-124 you mention the six-stage framework by Arksey & O' Malley and you cite a paper by Alan MMD et al which is not relevant. I can see that you have the reference from Arksey & O' Malley on number [36] but this is not cited appropriately.

--

Reviewer #3: General Comments

Thank you for the privilege to review this manuscript, titled “Use of telemedicine in general practice in Europe since the COVID-19 pandemic: a scoping review of patient and practitioner perspectives. 

The aim of this study was “to conduct a scoping review of the literature examining the use of telemedicine for delivering routine general practice care since the start of the pandemic from the perspectives of patients and practitioners”. Indeed, this is a timely and relevant piece of work, and the authors are commended for that effort. 

The authors used the six-stage framework developed by Arksey and O’Malley, blended with recommendations by Levac et al to review the existing literature on the subject matter. However, further works are needed on the explanation of the method used for this study. A brief description of the Arksey and O’ Malley framework will provide the reader with a better introduction to the framework, rather than leaving it to the reader to go and find out the paper discussing this framework. 

Specific Comments

This paper would need some revisions to allow for clarity and consistence to the reader. The details of which are indicated in the subsequent comments in this report. 

Major Comments

-----------------

Abstract

1. The abstract is fairly presented. However, in line 25 the word to be used is “adapt” instead of “adopt”.

2. In line 36, the correct word to use is “analyse” instead of “interpret” as the stated method is not used to interpret but to thematically analyse the data. 

3. In line 79, how are the authors defining a “bigger country”? They need to qualify this phrase.

4. In line 85, please use “At that time” instead of “At this time” and also replace “The” with “the”

Background

1. In line 61, add a coma after the word pandemic.

Methodology

1. Authors should state their justification of using a scoping review method instead of other available options. 

2. A brief description of the Arksey and O’ Malley framework will provide the reader with a better introduction of the framework, rather than leaving it to the reader to refer to the paper discussing this framework, before understanding this manuscript.

3. The search string strategy for the chosen database is better presented in a Table form rather than in a Figure form.

4. In the paragraph beginning at line 201, I recommend that authors list all the countries involved in the study, even though they are later shown in the data charting stage.

5. Why is the C (Critical) capitalized in line 206?

6. These are referred to as open ended questions. Please correct all entries accordingly.

7. In line 187, who are these experts? A clear definition is required as well as the selection criteria used to enroll these experts.

Results

1. In line 211, add “s” at the end of the word “practitioner”

Discussions

1. Under methodological challenges, please discuss also the expert sampling biases. 

2. Comparison with existing literature, in line 317 to 318, authors subscribe to a certain belief. Could you please explain the basis of that statement, by providing undisputed evidence? 

Reference

I am afraid, that the authors seem to lack attention to detail in matters of referencing, yet there are clear guidelines with examples for building up Plos One references. Almost 25% of the references have errors. Please follow the specific guidelines instructions and correct the identified errors accordingly. 

Ethics Statement

There is an omission of the study statement of ethics, indicated that the cohorts consented to the study.

Other Comments

1. Data Availability Statement, Authors provided the data availability statement that in not in line with Plos One requirements. They did not specify where the minimal data set underlying the results described in their manuscript can be found. This data should be available in Plos One recommended places such as public repository sites or provided as a supplementary file.

As a reviewer of this manuscript, I should be able to easily access this data set for verification of certain issues. Can the authors therefore, provide this minimal dataset?

2. Inconsistent placement of the full-stop (period) after the in-text citation number or before the in-text citation number. For example (1). Or For example. (1) I recommend you place the period after the citation number.

6. PLOS authors have the option to publish the peer review history of their article (what does this mean?). If published, this will include your full peer review and any attached files.

**Do you want your identity to be public for this peer review?** For information about this choice, including consent withdrawal, please see our Privacy Policy.

Reviewer #1: No

Reviewer #2: No

Reviewer #3: Yes: Benson Ncube

---

## [Editor Report · Decision Letter 1]

20 Sep 2023

PDIG-D-22-00363R1

Use of telemedicine in general practice in Europe since the COVID-19 pandemic: a scoping review of patient and practitioner perspectives.

PLOS Digital Health

Dear Dr. McCombe,

Thank you for revising the manuscript addressing reviewers' comments. I noticed a small typo in the new Author summary: GPs having to adopt... I think you mean: having to adapt. 

If you could please do a final check on this, and resubmit, we would be pleased to accept the paper for publication.

Please submit your revised manuscript within 30 days Oct 20 2023 11:59PM. If you will need more time than this to complete your revisions, please reply to this message or contact the journal office at digitalhealth@plos.org. Please include the following items when submitting your revised manuscript:

We look forward to receiving your revised manuscript.

Kind regards,

Valentina Lichtner

Academic Editor

PLOS Digital Health

Journal Requirements:

Additional Editor Comments (if provided):

Thank you for revising the manuscript addressing reviewers' comments. I noticed a small typo in the new Author summary: GPs having to adopt... I think you mean: having to adapt. 

If you could please check this, and resubmit, we would be pleased to accept the paper.
---

## [Editor Report · Decision Letter 2]

6 Dec 2023

Use of telemedicine in general practice in Europe since the COVID-19 pandemic: a scoping review of patient and practitioner perspectives.

PDIG-D-22-00363R2

Dear Dr McCombe,

We are pleased to inform you that your manuscript 'Use of telemedicine in general practice in Europe since the COVID-19 pandemic: a scoping review of patient and practitioner perspectives.' has been provisionally accepted for publication in PLOS Digital Health.

Best regards,

Valentina Lichtner

Academic Editor

PLOS Digital Health